# Multidimensional Biomechanics-Based Score to Assess Disease Progression in Duchenne Muscular Dystrophy

**DOI:** 10.3390/s23020831

**Published:** 2023-01-11

**Authors:** Carolina Migliorelli, Meritxell Gómez-Martinez, Paula Subías-Beltrán, Mireia Claramunt-Molet, Sebastian Idelsohn-Zielonka, Eudald Mas-Hurtado, Felip Miralles, Marisol Montolio, Marina Roselló-Ruano, Julita Medina-Cantillo

**Affiliations:** 1Unit of Digital Health, Eurecat, Centre Tecnològic de Catalunya, 08005 Barcelona, Spain; 2Ephion Health, 08005 Barcelona, Spain; 3Duchenne Parent Project, 28032 Madrid, Spain; 4Department of Cell Biology, Fisiology and Immunology, Faculty of Biology, University of Barcelona, 08007 Barcelona, Spain; 5Unidad de Patología Neuromuscular, Servicio de Rehabilitación, Hospital Sant Joan de Déu Barcelona, Passeig Sant Joan de Déu, 2, 08950 Esplugues de Llobregat, Spain; 6Investigación Aplicada en Enfermedades Neuromusculares, Institut de Recerca Sant Joan de Déu, Santa Rosa 39-57, 08950 Esplugues de Llobregat, Spain

**Keywords:** Duchenne muscular dystrophy, becker muscular dystrophy, six minute-walk test, north star ambulatory assessment, biomechanics, biomechanics data analysis, gait

## Abstract

(1) Background: Duchenne (DMD) is a rare neuromuscular disease that progressively weakens muscles, which severely impairs gait capacity. The Six Minute-Walk Test (6MWT), which is commonly used to evaluate and monitor the disease’s evolution, presents significant variability due to extrinsic factors such as patient motivation, fatigue, and learning effects. Therefore, there is a clear need for the establishment of precise clinical endpoints to measure patient mobility. (2) Methods: A novel score (6M+ and 2M+) is proposed, which is derived from the use of a new portable monitoring system capable of carrying out a complete gait analysis. The system includes several biomechanical sensors: a heart rate band, inertial measurement units, electromyography shorts, and plantar pressure insoles. The scores were obtained by processing the sensor signals and via gaussian-mixture clustering. (3) Results: The 6M+ and 2M+ scores were evaluated against the North Star Ambulatory Assessment (NSAA), the gold-standard for measuring DMD, and six- and two-minute distances. The 6M+ and 2M+ tests led to superior distances when tested against the NSAA. The 6M+ test and the 2M+ test in particular were the most correlated with age, suggesting that these scores better characterize the gait regressions in DMD. Additionally, the 2M+ test demonstrated an accuracy and stability similar to the 6M+ test. (4) Conclusions: The novel monitoring system described herein exhibited good usability with respect to functional testing in a clinical environment and demonstrated an improvement in the objectivity and reliability of monitoring the evolution of neuromuscular diseases.

## 1. Introduction

Duchenne Muscular Dystrophy (DMD) is a rare, progressive, neuromuscular disease affecting about one in five thousand newborn boys [1,2]. It occurs due to a gene mutation that inhibits the generation of the Dystrophin protein, thereby causing progressive muscle degeneration. The disease affects mostly males, symptoms usually start at the age of 2–3 years, and it is diagnosed at the age of 4 years, on average. The resulting muscle loss causes a progressive weakness that starts in the lower limbs and severely affects an individual’s gait capacity. Children usually need a wheelchair by the age of 15 years and have an average lifespan of about 30 years [3,4]. The Six Minute-Walk Test (6MWT) is a submaximal exercise test used to assess aerobic capacity and endurance that is universally adopted by clinicians to monitor and perform follow-ups regarding the evolution of patients. Although the test was first used to evaluate individuals with cardiopulmonary disorders, it has now been applied to assess numerous other diseases [5]. It is frequently used for DMD in clinical practice and research, with some modifications. The 6MWT involves measuring the distance that a patient can walk back and forth, as fast as possible and without running, within a corridor of normally 25 m in 6 min. For safety, the evaluator usually walks behind the patient in case of a fall. It has been demonstrated that the resulting distance, called the Six Minute-Walk Distance (6MWD), directly relates to the health status of a patient [6,7].

However, significant variability due to learning effects, motivation, fatigue, and daily variations is well-documented in the literature [8,9]. This variability is especially detrimental to DMD patients because the physical and/or cognitive state required to perform the six-minute walk can constitute a large effort, especially for boys at an early age. Another limitation of the 6MWT is that its results can be affected by a variety of factors, including age; sex; height; weight or nutritional status, and, more importantly, musculoskeletal problems; and cognitive function [10]. Additionally, there is conflicting evidence regarding the 6MWT’s measurement properties in relation to different chronic pediatric conditions [11]. As a result, changes detected by the 6MWT should be interpreted cautiously, with a consideration of whether they are significant for the particular patient. The Two-Minute-Walk Test (2MWT), a shorter version of the test, is especially recommended for pediatric populations with neuromuscular diseases, although it is not generally adapted clinically and shares limitations similar to the longer test [12,13,14]. There is a clear need for improved clinical endpoints that are unaffected by a user’s motivation or fatigue during the test to quantify the functional progression of the disease. These digital clinical endpoints would increase the quantity and quality of captured data, thereby allowing for the monitoring of the evolution of neuromuscular disorders with increased accuracy. New endpoints are especially relevant for the development of novel therapies wherein current endpoints obtained from the 6MWT and 2MWT offer insufficient sensitivity or quality to reveal statistically significant changes throughout a patient’s treatment [15,16]. Finally, there are no automatic systems on the market that allow for the remote measurement of the 6MWT and other digitalized endpoints. This fact can be used to design patient empowerment solutions. Therefore, a sensor’s output must be better evaluated in order to produce a trustworthy and functional system [5].

The current work proposes—with a patient-centric focus—the generation of a novel score named 6M+ and 2M+, which is derived from the measurement of a set of biomechanical and physiological variables collected as a part of a pilot test incorporating children suffering from DMD using a new monitoring system capable of performing a complete gait analysis integrated into a 6MWT standard protocol [17]. Data originating from multiple sensors were measured during the execution of the 6MWT and 2MWT. These measurements allowed for the extraction of a series of features that were subsequently combined using statistical analysis and unsupervised learning techniques (gaussian-mixture models) to obtain a novel score in order to correlate the progression of DMD in a more reliable manner. The developed 6M+ and 2M+ scores were evaluated against the North Star Ambulatory Assessment (NSAA) [18,19], an accepted gold-standard for measuring functional motor abilities, and against the 6MWD and 2MWD as well. The new platform offers a user-friendly, plug-and-play system that integrates different off-the-shelf biomechanical sensors and enables synchronous, remote, and automated data analysis.

## 2. Materials and Methods

The pilot study was conducted on 28 male children suffering from DMD (age 6.3–15.4 years) who attended control visits at the Rehabilitation Department of Sant Joan de Déu Hospital (Barcelona, Spain). Most of the patients were treated with corticoids. The control group consisted of 27 healthy male children (aged 9.0–13.9 years old) who attended Maristes la Inmaculada School (Barcelona). Tested subjects did not show any adverse reactions nor intolerances while using the systems. One subject felt uncomfortable wearing the insoles and another refused to use the EMG sensor.

### 2.1. Ethical Statements

The study was conducted in accordance with the Declaration of Helsinki and approved by the Institutional Ethics committee of Sant Joan de Déu Hospital (protocol code PS-07-19/PIC-120-19 and date of approval 27 June 2019) for studies involving humans. In addition, the study was approved by The Spanish Agency of Medicines and Medical Devices (AEMPS) (protocol code 772/19/EC).

### 2.2. Sensor and System Characteristics

During the test, multiple sensors were attached to different areas of the body to measure the biomechanics of the 6MWT. The types of attached sensors and their information is summarized in Table 1. The list of all acquired variables was grouped into five categories: Cardiac, EMG, Spatio-temporal, Kinematics, and Plantar pressure. The position of the sensors on the body is shown in Figure 1.

For the sEMG recordings, textile shorts with embedded conductive electrodes (dry electrodes made of silver-coated textile yarn) were used, which covered three main muscle groups: gluteus, hamstrings, and quadriceps. The best-fitting shorts were chosen for each participant. A small amount of water was applied to the electrodes to ensure adequate signal conduction, as previously recommended in [20]. EMG signals were captured at 1000 Hz; average rectified data were transmitted wirelessly at 25 Hz to a cellphone. Signal-processing methods and validation are covered in [21,22]. The purpose of this study with respect to sEMG was not to measure single muscle activity but a group of muscles. With regard to this general purpose, the reliability of the shorts, which facilitated the positioning of the sensors in a large area, was tested and validated in previous studies [23,24].

### 2.3. Performed Tests

The 6MWT was carried out in a standardized, level, undisturbed, and well-lit corridor that measured 25 m. Two cones were used to define the start and the end of the 25 m walkway. Prior to each test, surface electromyography (sEMG) signals were calibrated by having the participant perform a maximal voluntary contraction of the hamstrings, quadriceps, and gluteus muscles. Furthermore, the insoles were also calibrated. All calibration was performed according to a standard procedure developed using the Ephion Mobility [17] system. The participants were instructed to walk as fast as possible, without running, in a straight line back and forth between two cones for 6 min. If they needed to rest, they could lean against the wall. During these breaks, the stopwatch continued to run.

The completed distance was measured at two minutes (2MWD) and at six minutes (6MWD). The 6MWT test was conducted during three visits:Visit 1 (t0). The first visit, tagged as “t0”, was conducted to collect baseline data. During this visit, the participant performed the 6MWT once.Visit 2 (t0r). The objective of the second visit, which was labelled “t0r,” was to determine test–retest reliability. A subgroup of participants, made up of those who lived in the hospital area, underwent the test after a break of about 3–4 weeks.Visit 3 (t1). The third visit was tagged as “t1” and was planned in order to validate the test’s responsiveness. The test was performed with a time gap between t0 and t1 of approximately 6 months.

At each visit and for each patient, clinicians computed the NSAA, a 17-item scale used to rate functional motor abilities such as standing from a chair, walking, running, climbing stairs, etc. Control subjects received the maximum NSAA score (34). Figure 2-right shows the NSAA at the initial (t0), the second (t0r), and the third session (t1) for each subject. Subjects with medium NSAA values (from 15 to 20) at t0 suffered an abrupt decline between t0 and t1. 6MWD was also computed (Figure 2-left). Up to 6 patients showed differences in distances higher than 30 m between t0 and t0r, which is considered clinically significant [25]. NSAA was used for posterior comparison with the computed score as well as distance measurements (6MWD and 2MWD). The data analysis pipeline was also executed for both time-windows.

### 2.4. Data Analysis 

Figure 3 shows the pipeline with respect to obtaining a multidimensional score considering the information from the five categories. The pipeline is divided into one pre-processing stage and two processing stages. In the first processing stage, a unified score was generated, which was computed from the specific variables selected to characterize the pathology. This process of feature selection accounted for data points from each of the five categories. During the second stage, a “physiological gait model” (PGM) composed of the five categories was obtained to subsequently evaluate the scores for pathological subjects by analyzing the distance from subject to the PGM in the feature-space. This entire process was calculated for both 6 and 2 min, yielding scores of 6M+ and 2M+, respectively.

#### 2.4.1. Pre-Processing and Feature Extraction

The Annex I (Appendix A) lists and describes each calculated variable from the available sensors. These variables were selected according to a clinician’s assessment and the capabilities of the used devices.

Kinematics of trunk, hip, knee, and ankle; ground reaction forces; center-of-pressure evolution; and averaged rectified EMG activity of gluteus, hamstrings, and quadriceps were obtained during the 6MWT. EMG data were normalized to MVC and to the maximum value during the test. Data from the entire test were divided into gait cycles. A gait cycle describes a cyclical walking pattern; it starts when one foot contacts the ground and ends when the same foot contacts the ground again. The subjects’ gait patterns were then determined by taking the mean of all gait cycles per each parameter. Thus, it was possible to calculate numerous characteristics of the curves, including their maximum, minimum, peak, value at contact, and value at take-off. The mean was also calculated for the first and last two minutes and the first and last two laps.

Spatio-temporal parameters were calculated from the inertial sensors and plantar pressure data. Inertial sensor attached to the trunk was used to identify the turns; then, distance was calculated as the total number of turns multiplied by the length of the corridor (25 m). Velocity in each turn was calculated by dividing the length of the corridor by the time taken to cover the corridor’s distance. Number of steps were calculated using ground reaction force data as well as the different phases in the gait cycle reported in percentages (stance, swing, single support, double support, pressing, and loading). Step length and step height were calculated using insole inertial sensor as explained in [26]. HR and RR values were obtained directly from the sensor every second during the whole test. From those recorded signals, the evolution for the HR and RR during the test was calculated, including different parameters such as slope, variability, and difference between initial and final values. 

Anatomical joint angles were calculated from gyroscope data of the sensors attached to the consecutive segments. A method based on integrating angular velocity [27] and the correction of the drift was used to compute absolute segment angles [28]. To obtain the relative anatomical joint angles, motion segment angle was subtracted from the reference segment angle. Trunk segment was used as the reference body segment.

#### 2.4.2. First Stage

At this step, the variables that were irrelevant for differentiating between normal and pathological behavior were eliminated; it consists of the following sub-stages:Data cleansing. DMD disorder is considered to affect the right and left limbs almost symmetrically [29]. Thus, the variables corresponding to the sensors placed in both right and left parts of the body were averaged into a single measure.Feature selection I. Robust variable selection. The most robust and repeatable variables were selected by comparing t0 and t0r sessions for all subjects. These are expected to have similar values between sessions as they were recorded in a short time-window. As the sets of data are used to compare an increase from the same individuals at different sessions, the data are paired. To determine which statistical test was the most suitable for each variable, a Shapiro–Wilk test was applied. If the variable followed a normal distribution for both sessions, a Paired Samples T-Test was applied. If not, the chosen test was the Wilcoxon Signed-Rank Test. The variables that exhibited the same underlying distribution for both sessions (*p* > 0.05) were considered robust and used for subsequent analysis.Feature selection II. Selection of the variables that best characterize the pathology. This step compared how the control and pathological distributions differ between each other. For each one of the variables, the Kruskal–Wallis test for unpaired data was used to assess if control and pathological distributions were significantly different. The values that showed a significant difference (*p* < 0.01) were selected for further analysis.Generation of a unified score for each of the categories. The selected variables, which belonged to one of the five categories, were first normalized (the minimum value was set to 0 and the maximum value to 1). The direction of each variable had to be the same. In other words, it was necessary that the pathological cases always showed the lowest numbers and the control cases the highest. Therefore, if the median value of control cases was lower than the pathological median value, the variable was inverted. Finally, a single variable for each category was computed as the mean of the normalized variables belonging to that category and adjusted between 0 and 100.

#### 2.4.3. Second Stage

While determining a category-by-category score may be useful to assess a subject’s performance in terms of different aspects of their gait, significant insights can be obtained from the definition of a unique score that characterizes the status of the subject at the moment of the evaluation. This stage determined the physiological gait model and subsequent score for each subject. It consisted of the following sub-stages:Generation of the physiological gait model. For the control group, the computed scores for each category (Spatio-temporal, Cardiac, Kinematics, EMG, and Plantar pressure scores) were used to define a 5-dimensional gaussian-mixture model, where each dimension corresponded to one category. This model was defined as the physiological gait model (PGM). By using the Bayesian Information Criterion [30], the number of components for modelling the PGM as well as the covariance type (from spherical, diagonal, tied, and full) were determined.Obtaining the global score for controls pathological observations. Both the pathological and control subjects achieved a score for both the 6 min (6M+) and 2 min (2M+) tests. These scores were calculated as the weighted log probability of belonging to the PGM and, subsequently, scaled from 0 to 100. The highest score was 100, which was expected for control subjects. The further an observation was from the model, the lower was its score. Indeed, controls would score better since they are predicted to have a higher likelihood of fitting PGM. Depending on their severity, pathological individuals were predicted to gradually stray from the PGM. Consequently, they are expected to gradually receive lower scores.

### 2.5. Score Validation

The computed pathology score (6M+ and 2M+ for 6 and 2 min, respectively) was compared to the normalized NSAA score (from 0 to 100). The 6MWD and 2MWD scores were also normalized and compared with NSAA. For the four measurements, discrepancies were computed as follows: 100 * (1 − |score_p_ − score_v_|)(1)
where score_p_ and score_v_ represent the predicted score and the NSAA-normalized score, respectively.

As DMD precipitates progressive muscle degeneration, mobility performance is expected to continuously decrease with age. The linear regression between severity—assessed through the different scores (NSAA, 2M+, 6M+, 6MWD and 2MWD)—and age was computed. As mobility performance decreases with age, this correlation is supposed to be negative. A greater correlation with disease progression is expected for higher R^2^.

The consistency of the scores determined by test and retest is another factor to consider. For each subject who completed the test and retest sessions, the relative difference (obtained as percentual variation) between both sessions was computed for 6M+, 2M+, 6MWD, and 2MWD.

## 3. Results

### 3.1. Generation of the Multidimensional Score

The measurements of the five categories were assessed for 2 and 6 min intervals, thereby obtaining a score for each category and for each period. The categorical scores represent deviations of the pathological subjects in the five domains compared to a normal gait. For each category, a score of 100 indicates that the domain is not affected, that is, the subject behaves like the control population. Figure 4 shows the distribution of the scores for each of the categorized variables for both groups.

As an example, Figure 5 shows how different subjects presented variations in performances for the evaluated categories.

The multidimensional score enables a specific assessment of a disease’s progression for each category, allowing for the personalization of therapies according to those aspects for which the patients have more difficulty. Figure 6 shows the average evolution of the scores of each of the categories for the decreasing ranks of NSAA. The scores do not change linearly over categories as the disease progresses, showing their declines at different stages of the illness.

### 3.2. Generation of the 6M+ and 2M+ Scores

The PGM was generated using a gaussian-mixture model that combined the multicategory scores for the control observations. With regard to the modelling of the physiological data, the Bayesian Information Criterion determined four and five components for the 6M+ and 2M+ scores, respectively, and a diagonal covariance type for both scores.

### 3.3. The Pathological 2M+ and 6M+ Score and Its Comparison with NSAA and Distances

The NSAA scale was taken into consideration as the gold-standard. Figure 7 shows the global differences (discrepancies, resulting from computing Equation (1)) between the pathological scores (2M+ and 6M+) and 6MWD and 2MWD in comparison with NSAA). The 6M+ and 2M+ scores represented a higher degree of similarity to the NSAA for the controls compared to the 6MWD and 2MWD tests. For the DMD group, the 6M+ and 6MWD performances were similar (86.00 and 86.30 respectively), while 2M+ was slightly higher than 2MWD (85.03 and 79.68 respectively).

The performance of the newly proposed score was also evaluated for different NSAA ranges for the DMD subjects (Figure 8). For the evaluated ranges, a decrease followed by an increase in agreement with NSAA was observed, with the worst performances in the 15–19 NSAA scores.

The regression with the ages of the patients is shown in Figure 9. As a progressive disorder, it is expected to have a negative correlation with age. The 6M+ and 2M+ scores were the most negatively correlated with age (with R^2^ values of 0.41 and 0.49, respectively), with higher values than the NSAA score.

Finally, the relative difference between the test (t0) and retest (t0r) sessions was computed for 6M+, 2M+, 6MWD, and 2MWD as a percentual variation. The differences for 6M+ and 2M+ (expressed in mean ± SD percentages) were 3.64 ± 5.82% and 5.95 ± 6.96%, respectively, and the differences for 6MWD and 2MWD were 7.14 ± 6.92% and 10.50 ± 11.37%, respectively. 6M+ was lower than 6MWD (Wilcoxon signed-rank test, *p*-value = 0.039) and 2MWD (Wilcoxon signed-rank test, *p*-value = 0.013). 2M+ was lower than 6MWD (Wilcoxon signed-rank test; *p*-value = 0.101) and lower than 2MWD (Wilcoxon signed-rank test; *p*-value = 0.045).

## 4. Discussion

DMD produces progressive muscle degeneration that forces patients to compensate their natural movements with unnatural ones to maintain their ambulatory ability. This fact, combined with inequalities regarding the disease’s progression among different patients, produces different gait patterns and alterations that are reflected in the analyzed categories. Due to the multicategory score, wherein each category was assessed independently, it was possible to objectively monitor the evolution of the disease within each of the five categories (Figure 6). The multicategory score gives an in-depth insight of the disease’s progression to the clinician and allows for personalized treatment or therapy that specifically targets each category. This reduces the psychological pressure that patients may feel as they know that the global results of the test could affect their near and mid-term futures, and since these test results will be used to adapt treatment or even act as an excluding criterion in a patient’s eligibility to participate in a Clinical Trial.

In general, as the disease progressed and the scores decreased, different trends were found in the analyzed variables (Figure 6). For example, heart rate increased at rest, muscle load diminished, and there was a minor activation of quadriceps during the test due to muscle weakening. Hip flexion and knee flexion in the stance phase were lower for the most affected patients, and the knee was not flexed during the stances for the most severe cases. Toe–foot contact increased, with the severity modifying the shape of the ground vertical force (first a flatter and then less-prominent first peak). Finally, most of the spatio-temporal parameters measured as cadence, two-minute distance, step height, step length, stride velocity, and velocity decreased as the disease progresses. These findings agree with previously described results and demonstrate that the multicategory score provides greater sensitivity with respect to continuous changes occurring during the natural progression of the disease [31].

However, to analyze the progression of the pathology or measure the effectiveness of a pharmacological or rehabilitation therapy, it is convenient to obtain a single score per subject. Two individual scores, 6M+ and 2M+, were obtained and subsequently tested against the NSAA. The computed Scores outperformed the current scores (6MWD and 2MWD) based only on the distance (Figure 7). In addition, when assessing the robustness of the score in contrast with the 6MWD between the initial test (t0) and the retest (t0r), the relative degrees of variability of the 6M+ and 2M+ were significantly lower than those of the 6MWD and 2MWD, respectively. This observation supports the hypothesis that some biomechanical variables are less affected even when people exhibit variations in their performances as a result of subjective circumstances.

Although the NSAA might be considered a gold-standard, one of its main limitations is that the tested subjects present a sharp decrease between middle values. This effect is observed in the NSAA distribution (Figure 2-left), where only six patients in t0 or t1 sessions scored a NSAA between 10 to 20. This can be explained because the measurement method is based on the punctuation of different activities, wherein some of them are directly related and independent for each leg, whereas these pathologies are mainly symmetrical [32]. Accordingly, the 6M+ and 2M+ scores show the highest discrepancies with respect to these ranges of NSAA values (15–19 and 10–14) (Figure 8). To assess the sensitivity of the different metrics in terms of measuring the progression of the disease, the regression with age for all the measured scores was calculated (Figure 9). As DMD is a degenerative disorder, a negative regression was expected. The 6M+ test and the 2M+ test in particular were the most correlated with age, even outperforming the gold-standard NSAA score. These results suggest that the new score better characterizes the regressions in gait observed in DMD.

Performing a 6MWT requires an enormous effort for DMD patients, especially for the most severe cases; shorter versions, although less implemented clinically, such as 2MWT, have been demonstrated to correlate well with longer tests, but they both maintain reliability and responsiveness issues [8,9,10,11,12,13,14]. The 2M+ score demonstrated an accuracy and stability similar to the 6M+ score, greater than the 6MWD, and even greater than the 2MWD (Figure 7), while maintaining the independency of the system to external factors such as patients’ mood or willingness to perform the test. These results highlight the possibility of shortening the duration of the 6MWT to a 2MWT, thereby reducing the additional effort for the patients, a trait that is especially relevant for child populations.

Future work includes the adaptation of the system and the physiological gait model to other pathologies that affect mobility. Additional sensors could be added to the system to complement the current ones or be placed on other parts of the body to sense different types of activities. Moreover, future research lines will involve the use of this technology for the identification and employment of digital endpoints in clinical trials. 

## 5. Conclusions

The novel health-monitoring system Ephion Mobility provides a unique single score by using biomechanical and physiological data captured while conducting a 6MWT standard protocol. The system showed good usability in terms of functional testing in a clinical environment. This new solution demonstrated an improvement in objectivity and reliability with respect to monitoring the evolution of neuromuscular diseases. The possibility of splitting the score into different categories eases treatment personalization. Finally, the enhanced sensitivity of this resulting score will allow clinicians to shorten the current 6MWT (into a 2MWT), thereby reducing the effort required by the patient without losing important information.

## 6. Patents

A patent has been presented and it is currently under evaluation. 

Title: SYSTEM, METHOD, AND COMPUTER PROGRAMS FOR ASSESSMENT OF BODY MOVEMENT’S CONDITIONS OR DISORDERS 

Applicant: FUNDACIÓ EURECAT

## Figures and Tables

**Figure 1 sensors-23-00831-f001:**
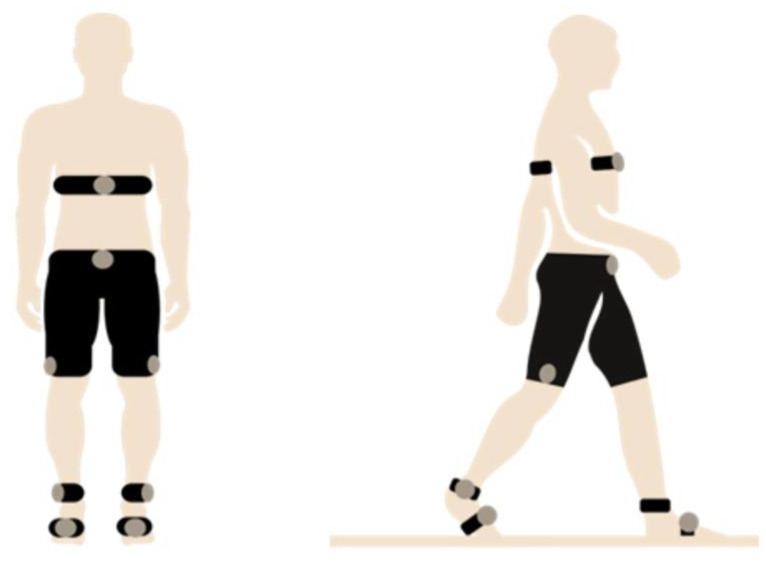
Position of sensors during the test execution.

**Figure 2 sensors-23-00831-f002:**
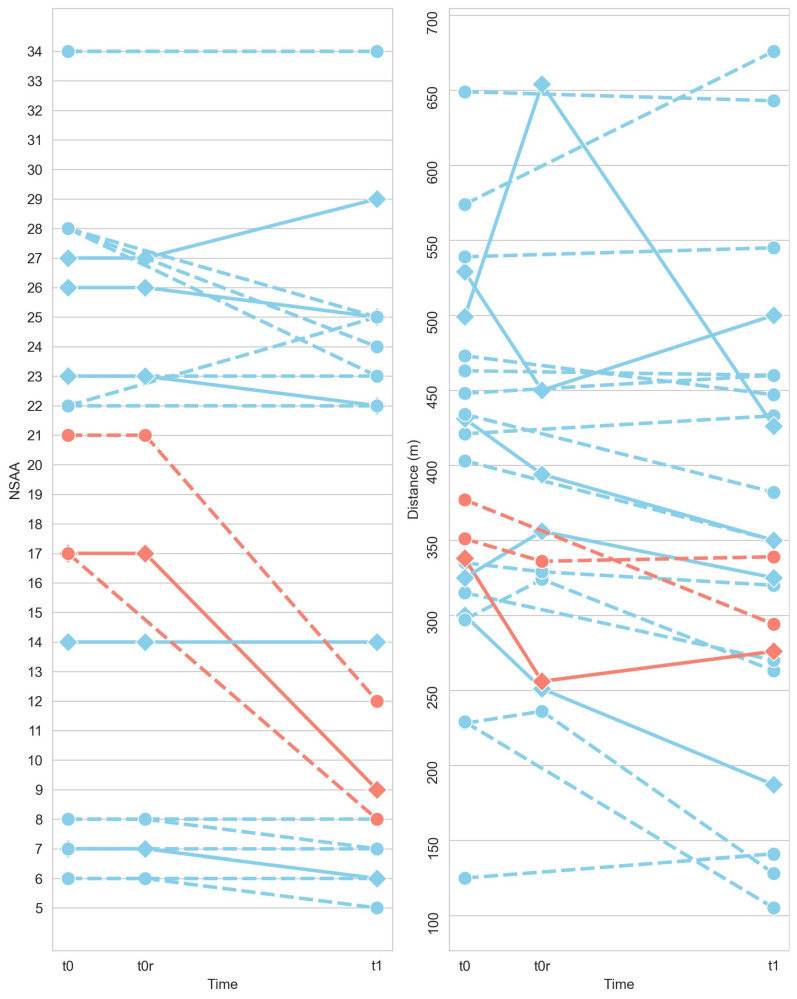
NSAA (**left**) and 6MWD (**right**) individual evolution for DMD (pathological) subjects between initial session (t0), the retest session (t0r), and the final session (6 months later, t1). Subjects colored in pink showed an abrupt decline in NSAA score between t0 and t1. Subjects with a solid line achieved a difference in distance between t0 and t0r greater than 30 m.

**Figure 3 sensors-23-00831-f003:**
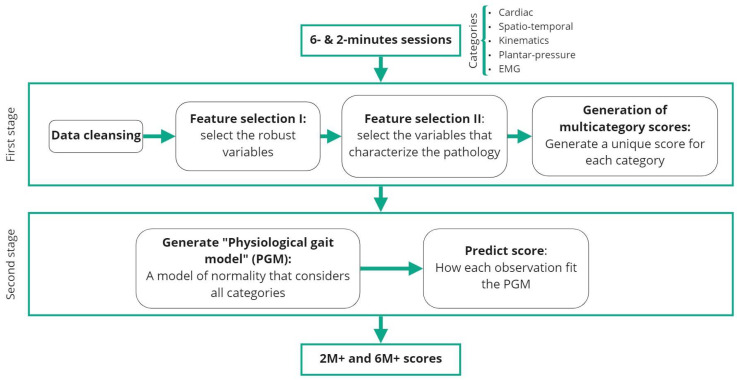
Pipeline for obtaining the global score.

**Figure 4 sensors-23-00831-f004:**
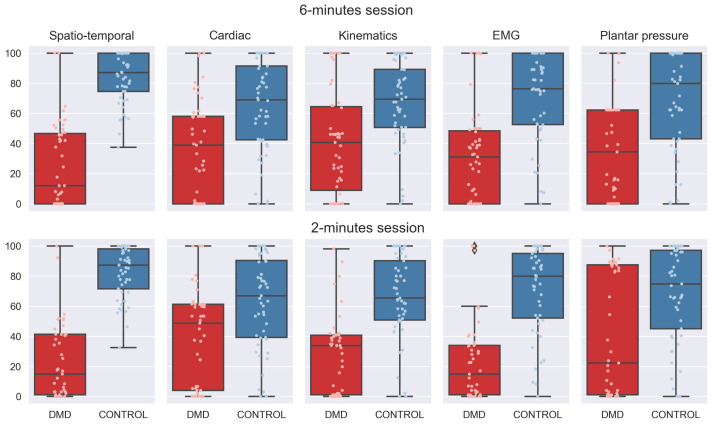
Boxplot distribution for controls (blue) and DMD-affected individuals (red) for each of the scores for the assessed categories, both for 6 and 2 min sessions. Control group achieved scores closer to 100 while DMD group achieved lower scores.

**Figure 5 sensors-23-00831-f005:**
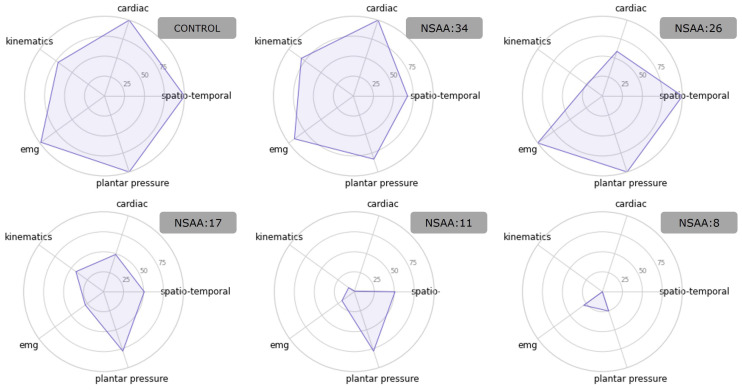
Spider plots of the predicted scores for a control subject and 5 DMD patients with different levels of affliction. As the NSAA scores decrease, so do the scores. However, these reductions are not uniform among categories, being highly dependent on the subjects.

**Figure 6 sensors-23-00831-f006:**
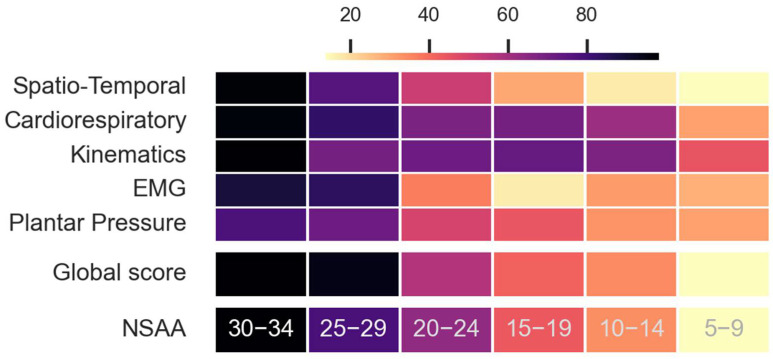
Progression of the multicategory score and global score for different NSAA severities. The color bar represents the normalized averaged scores for each one of the NSAA ranges, as well as for the normalized NSAA. For high NSAA values (30–34), scores are the highest and progressively decrease until lower NSAA values (5–9). All of these decreases are not linear, indicating that each category is impacted at a distinct stage of the disease’s development.

**Figure 7 sensors-23-00831-f007:**
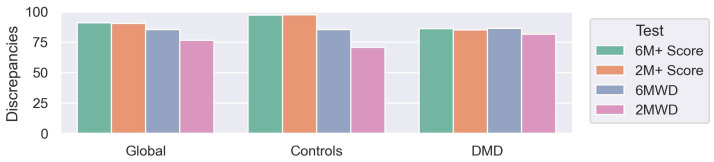
Discrepancies of 6 and 2 min scores (6M+ and 2M+, respectively), 6MWT, and 2MWT in comparison with NSAA score, computed as depicted in Equation (1).

**Figure 8 sensors-23-00831-f008:**
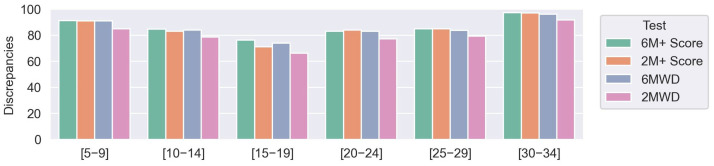
Discrepancies for different NSAA ranges, for 6 and 2 min scores (6M+ and 2M+, respectively), 6MWT, and 2MWT in comparison with NSAA score, computed as depicted in Equation (1).

**Figure 9 sensors-23-00831-f009:**
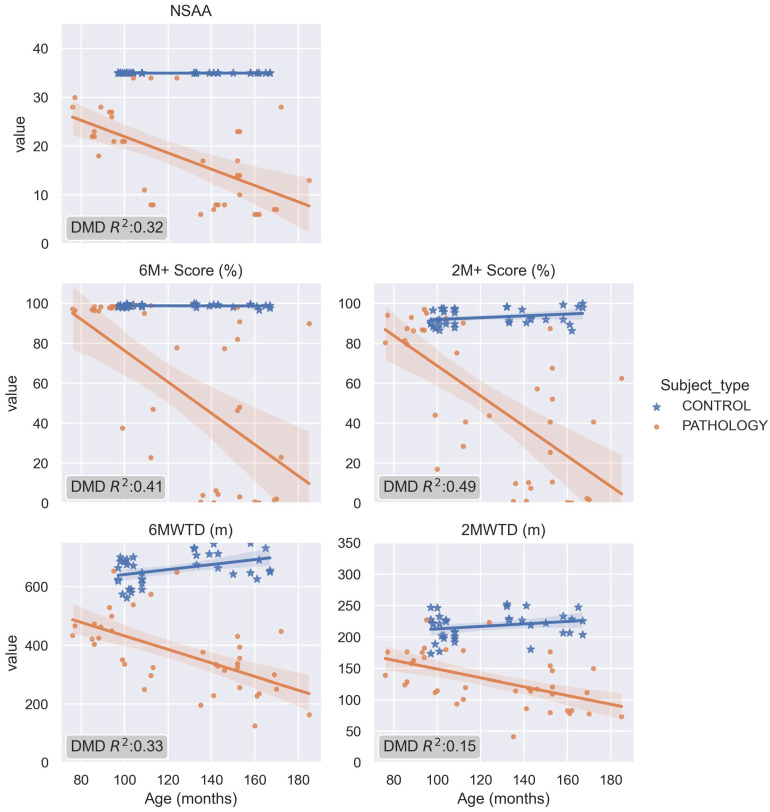
Regression of the target variables from each score methodology with age.

**Table 1 sensors-23-00831-t001:** List of the different wearable sensors that were attached to the patient during the test.

Sensor Type	Description	Device Used	Fs	Other Specifications	Category
Surface EMGs	EMGs were integrated in custom-made, size-appropriate shorts that enabled recording of muscle activity.	Mshorts Myontec	1000 Hz captured 25 Hz transmitted	24-bit ADC, sampling rate of 1000 Hz, bandwidth of 40–200 Hz (−3 dB) 25 Hz (rectified and averaged from 1000 Hz Raw EMG) Electrodes: silver-coated yarn	EMG
Inertial sensors	Sensors were placed at the thigh, shank, and in the plantar insoles to measure joint kinematics.	Movesense inertial sensors	104 Hz	Gyroscope: ±2000 dps Accelerometer: ±8 g	Spatio-temporal, kinematics
Heart rate	A strap was applied below the chest to measure cardiorespiratory parameters.	Movesense HR sensor	1 Hz		Cardiac
Plantar pressure	Pressure sensors were integrated in insoles, which were size-appropriate and used to obtain information on weight distribution during gait.	Moticon plantar insoles	100 Hz	Gyroscope: ±2000 dps Accelerometer: ±16 g	Plantar pressure, spatio-temporal, kinematics

## Data Availability

The conditions of our ethical approval do not allow for the public archiving of anonymized patient data. Derived data supporting the findings are available on reasonable request from the corresponding author.

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
