# Peer review of "Multidimensional Biomechanics-Based Score to Assess Disease Progression in Duchenne Muscular Dystrophy"

_sensors, 2023, doi:10.3390/s23020831_

Round 1

Reviewer 1 Report

The authors describe a system for improving the monitoring of patients with DD. The aim of the paper is well described and the need justified. However, the methods and results are confusing, and they need to improve these sections before accepting this work.

Specific comments:

format:

-         Please review and correct sentences in lines 59-62, 110, 115-118 and 131-132

-          check the formatting of paragraph in lines 158-164

contents

-  In line 82 it says that you used machine learning techniques, but they are not described nor mentioned in the rest of the manuscript

- While the need of more precise clinical endpoints is exposed, it is not clear why the apparatus will eliminate this bias: for example, the musculoskeletal condition and fatigue are two of them, and for sure they will affect the measurements with the proposed system.

- line 104-106. How you selected the location of the sEMG electrodes? Please include a description of the electrodes, number of channels, recording method, sampling frequency etc. How do you know the electrodes were localized in the hamstrings or other muscles? what is the reliability of this position for the second/third tests and how did you measure it?

- caption of figure 1 does not seem to correspond to figure 1. For instance, there are no plots labelled A, B, C, or D. There are only two plots and they are not labelled. In addition, there are no solid lined showing subjects that “achieved a difference of distance between t0 and t0r greater than 30 meters". Also it is not clear if the plot is showing controls, patients or both

- lines 158-154: Please add a description of the system/ sensor used to measure the EMG signals. How can you be sure that you were in fact measuring the activities of the described muscles and not of neighbor muscles? There is a big body of literature supporting the need to correctly localize the EMG sensors for getting repeatable outcomes, and for me this is not easy to achieve using prebuild shorts (with embedded EMG electrodes). More likely is that you are measuring the activity of some body segments that may or may not be the targeted muscles you are describing. I’m not saying that the measurements are not valid, but they are not clearly related to the activity of a given muscle. Also, I found that you need to describe better the sensors you are using. For instance, the EMG sensors in terms of area, recording modality (bipolar or monopolar), if they are dry electrodes, etc. The same with other sensors (ECG, inertial, etc.) and their locations. When you say “The subject pattern was then determined by taking the mean of all gait cycles” are you referring to the muscular pattern? In terms of what? Amplitude of the signal? Timing?

Lines 165-168

Please describe the variables. For instance, how do you obtain the velocity? What about the HR, are you referring to the slope of the R segments or something? It is not clear. This applies for all the variables in the study. It says that there is an Annex I, but I do not have it with the report and in any case this information is very sensible for understanding the methods you are applying, so I think it is better to include them as a section in the main paper.

Lines 186-187. When you apply the kruskall- wallis test, you are comparing the behavior between groups for every trial independently, right? Why not applying a Friedmann test to analyze the evolution of both groups (like repeated measures)

Lines 191-196. Why there is need for inverting the variable? Not clear

Lines 203-205. Which merged variable? Please explain

Eq 1. How do you obtain scorep and scorev ? Is scorev obtained somehow from the NSAA score, or is the actual NSAA? How do you obtain the model of normality in fig. 1? Are you referring to a regression model? If so, which variables are you considering and how you attribute the probability of the scores that a given individual obtained? Not clear

Figure 3. I am guessing that the control plot is depicted in blue, but this information should be given either in the caption or in the figure. Regarding the box plots, were the upper limits forced to 100? It is odd that the upper-limits are 100 for all the boxes… 

I found Figures 5-7 hard to read. I am not sure if they are well described. For instance: discrepancies in figure 6 are actual differences (as a subtraction) with a some normalized form of the NSAA? In fig 5 does the x-axis refer to the time-evolution of the indexes? What do the colors represent, and why the NSAA have different colors if is the variable of reference? In addition, please correct the term Emg to EMG. Please include a more detailed description of the figures so they can be more easily interpreted. For me, it is hard to draw conclusions from them.

The discussion is interesting, but the methods need to be described more clearly and the presentation of the results must be enhanced to better understand them and support the claims in this section. I see the point in obtaining a score that is based on a 2 minutes’ test for these patients, but I do not see how the described system aids in controlling “factors like patients’ mood or willingness to perform the test” (lines 322-323), it only makes it easier by shortening the test time. Therefore, the presented solution may be not addressing such factors and conclusions regarding this fact should be revised.

Author Response

We appreciate the reviewer comments. He have uploaded the responses in a separate word file. 

Reviewer 2 Report

A brief summary:

The six minute walking test (6MWT), which was developed by the American Thoracic Society, is well-known as an exercise test to assess aerobic capacity and endurance. This paper proposed two novel scores (6M+ and 2M+) as performance indices, to assess disease progression on Duchenne Muscular Dystrophy (DMD). With the integration of the biomechanical sensors into the traditional 6MWT, the scores can be calculated from the acquired data. The scores of 6M+ and 2M+, were evaluated in comparison to North Star Ambulatory Assessment(NSAA), and six and two-minute distances (6MWD and 2MWD). Since this paper was submitted to the special issue titled “Sensor Technologies for Gait Analysis”, how to utilize the biomechanical sensors and the acquired data during gait is of the readers’ concern. However, technical details are not sufficient in this manucript.

General concept comments:

The paper needs revision and my comments are as follows.

1. The specifications of the used biomechanical sensors such as the manufacturer, sampling rate, and accuracy were not found in this manuscript, which are associated with the data quality. The reviewer strongly suggest the authors read the papers published in the same issue of Sensors. More details on these sensors should be given.

2. Could the authors show us the locations of the biomechanical sensors and the experimental setup? More figures are desired, instead of the descriptions listed in Table 1.

3. The authors mentioned that the joint kinematics during gait were calculated from the inertial sensors. Please introduce the algorithm or give some related citations.

4. The paper mainly analyze several different scores, including the proposed 6M+, 2M+, NSAA, 6MWD and 2MWD. In the paragraph of “Feature selection I”(Page 6), the author mentioned that a Shapiro-Wilk test was applied. According to the results of Shapiro-Wilk test, were all the variables normal distributed or not? From the Boxplot distribution (Fig. 3), the data seem to be non-Gaussian distributed.

5. Similar to Fig. 3, the distribution of the scores for each one of the categorized variables for both groups for 2 minutes should also be given.

Author Response

We appreciate the review. We have uploaded a word document adressing all the reviewer's points.

Round 2

Reviewer 2 Report

Please check the uploaded file.

Author Response

Please, find attached the reviewer answer in the Word document. 

Round 3

Reviewer 2 Report

I have checked the revised manuscript and coverletter. Since the authors carefully explained what they've done with respect to the calculation of joint angles. I think the submitted paper can be accepted at present.

Author Response

We thank the reviewer for their review. We believe that these comments have improved the manuscript and its intelligibility.